# Rcs Phosphorelay Responses to Truncated Lipopolysaccharide-Induced Cell Envelope Stress in *Yersinia enterocolitica*

**DOI:** 10.3390/molecules25235718

**Published:** 2020-12-03

**Authors:** Jiao Meng, Junhong Xu, Can Huang, Jingyu Chen

**Affiliations:** Beijing Laboratory for Food Quality and Safety, College of Food Science & Nutritional Engineering, China Agricultural University, Beijing 100083, China; jiaomeng123@yeah.net (J.M.); xujunhong@cau.edu.cn (J.X.); canhuang_cau@hotmail.com (C.H.)

**Keywords:** *Yersinia enterocolitica*, lipopolysaccharide, envelope stress, Rcs phosphorelay

## Abstract

Lipopolysaccharide (LPS) is the major component of the outer membrane of Gram-negative bacteria, and its integrity is monitored by various stress response systems. Although the Rcs system is involved in the envelope stress response and regulates genes controlling numerous bacterial cell functions of *Yersinia enterocolitica*, whether it can sense the truncated LPS in *Y. enterocolitica* remains unclear. In this study, the deletion of the *Y. enterocolitica waaF* gene truncated the structure of LPS and produced a deep rough LPS. The truncated LPS increased the cell surface hydrophobicity and outer membrane permeability, generating cell envelope stress. The truncated LPS also directly exposed the smooth outer membrane to the external environment and attenuated the resistance to adverse conditions, such as impaired survival under polymyxin B and sodium dodecyl sulfate (SDS) exposure. Further phenotypic experiment and gene expression analysis indicated that the truncated LPS was correlated with the activation of the Rcs phosphorelay, thereby repressing cell motility and biofilm formation. Our findings highlight the importance of LPS integrity in maintaining membrane function and broaden the understanding of Rcs phosphorelay signaling in response to cell envelope stress, thus opening new avenues to develop effective antimicrobial agents for combating *Y. enterocolitica* infections.

## 1. Introduction

*Yersinia enterocolitica* is primarily a zoonotic pathogen frequently associated with human non-specific gastroenteritis [1,2,3]. *Y. enterocolitica* is listed in the annual reports of the European Food Safety Authority as the third most common enteropathogen, after *Campylobacter* and *Salmonella* [4]. In some countries, it is on par with *Salmonella* as a foodborne pathogen [5]. *Y. enterocolitica* can flourish at refrigeration temperatures and survive in different harsh environmental conditions, rendering it an important pathogen associated with foodborne infections [6]. *Y. enterocolitica* is widely distributed in the environment and can be found in soil, water, animals, and various food products [7]. *Y. enterocolitica* can invade the intestinal mucosa, multiplying and replicating within a phagocytic vacuole [8]. *Y. enterocolitica* is exposed to a range of envelope stresses due to both their environment and infectious nature [9]. Encountered stresses include desiccation, changing osmotic stress, temperature fluctuations, and exposure to cationic antimicrobial peptides [8,9]. Therefore, *Y. enterocolitica* transitions through a variety of environments and must respond to these envelope stresses with appropriate gene expression regulation to promote survival and growth within host tissues [9,10].

The Rcs (regulator of capsule synthesis) phosphorelay system, a non-orthodox two-component regulatory system found in many members of *Enterobacterales*, is one of the envelope stress response pathways [11,12]. The Rcs system can sense envelope damage or defects and regulate the transcriptome to relieve stress, which is particularly important for the survival and virulence of pathogenic bacteria [13]. The Rcs system is composed of three core proteins, the transmembrane sensor kinase RcsC, the transmembrane protein RcsD, and the response regulator RcsB [8,11,12]. In addition, an outer membrane-associated lipoprotein RcsF is required for the perception of many of the signals that have been shown to activate the Rcs phosphorelay [14]. In the signal transduction of the Rcs phosphorelay, RcsF senses the envelope stress signals and transmits it to RcsC [15]. With the participation of ATP, the hybrid sensor RcsC autophosphorylates at the conserved histidine residue His479 on its histidine kinase (HK) domain. Then, the phosphoryl group is transferred to the aspartate residue Asp875 on the phosphoryl receiver (PR) domain of RcsC. The phosphoryl group is subsequently transferred to the histidine residue His842 on the histidine-containing phosphotransmitter (HPT) domain of RcsD and finally to the aspartate residue Asp56 on the PR domain of RcsB [11,12]. RcsB can form homodimers or heterodimers with auxiliary proteins, such as RcsA, BglJ, MatA, GadE, and then bind to the conserved motif in Rcs-regulated genes to activate or inhibit their transcription [14,16,17,18]. In general, the Rcs system is activated by outer membrane damage, lipopolysaccharide (LPS) synthesis defects, peptidoglycan perturbation, and lipoprotein mislocalization, which then regulates genes involved in capsule biosynthesis, motility, biofilm formation, and virulence [12].

LPS is an essential structural molecule in the outermost part of the cell envelope, and it consists of three parts: lipid A, core polysaccharide, and O-specific polysaccharide (O-antigen) [19,20]. LPS is a physical barrier that can protect bacteria from the surrounding environment and is recognized by the immune system as a marker for detecting the invasion of bacterial pathogens responsible for the development of inflammatory reactions, and in extreme cases, it is resistant to internal toxic shock [21]. In addition to the role of the O-antigen in bioactivity, the structural integrity of LPS has important implications for bacterial survival and virulence, especially the relatively structurally conserved core polysaccharide [22,23]. Furthermore, defects in LPS biosynthesis can induce cell envelope stress, activating the Rcs system [24,25]. It has been reported that LPS defects impaired motility by repressing flagella gene expression through the Rcs signaling pathway in *Escherichia coli*, and only mutants with large LPS truncations showed significant defects in those behaviors [26]. Similarly, in a series of mutant strains of *E. coli* K12 that knocked out the core polysaccharide transferase gene in the *waa* gene cluster, Ren and colleagues found that only the Δ*waaF* strain might be sensed by the Rcs phosphorelay, leading to the production of colanic acid [23]. Although the Rcs system is involved in envelope stress responses and regulates various physiological behaviors of *Y. enterocolitica* [8,10], whether it can sense the truncated LPS in *Y. enterocolitica* is still unclear.

In this study, we showed that the deletion of the *Y. enterocolitica waaF* gene truncated the structure of LPS and generated a deep rough LPS. The truncated LPS increased the cell surface hydrophobicity and outer membrane permeability, generating cell envelope stress. This defect in LPS also caused changes in membrane surface smoothness and increased the sensitivity of *Y. enterocolitica* to environmental stress. The phenotypic experiment and gene expression analysis indicated that the truncated LPS was correlated with the activation of the Rcs phosphorelay, thereby repressing cell motility and biofilm formation. These findings reveal the importance of LPS integrity in maintaining membrane function and further deepen the understanding of Rcs phosphorelay in response to cell envelope stress in *Y. enterocolitica*.

## 2. Results

### 2.1. waaF Gene Knockout in Y. enterocolitica Constructed a Truncated LPS Structure

In order to obtain a truncated LPS, the core polysaccharide transferase gene *waaF* was knocked out in the chromosome of wild-type *Y. enterocolitica* (biotype 1B and serotype O:8). It has been reported that core polysaccharides of LPS are synthesized under the corresponding glycosyltransferases and phosphoribosyltransferases encoded by the *waa* gene cluster [27,28]. As each glycosyltransferase is highly substrate specific and temporally sequential, the absence of the *waaF* gene fails the synthesis of the encoded heptyl glycosyltransferase II, thus leading to a truncated LPS structure [23]. Theoretically, the mutated LPS caused by *waaF* deletion has only lipids and a portion of the inner core polysaccharides without the outer core polysaccharides and O-antigen, which is called deep rough LPS [22]. In this study, we found that the colony morphology of the Δ*waaF* strain had S-R (smooth to rough) variation caused by the *waaF* deletion (Figure 1A). This may be due to the absence of the LPS O-antigen structure in the Δ*waaF* mutant strain. Furthermore, silver-stained SDS-PAGE showed that most of the sugar chains were deleted from LPS in the Δ*waaF* mutant strain (Figure 1B), which directly verified that the deletion of *waaF* led to the formation of a truncated LPS structure in *Y. enterocolitica*.

### 2.2. Truncated LPS-Induced Envelope Stress in ΔwaaF Strain

LPS is the central structural and functional unit of the outer membrane of Gram-negative bacteria and plays a crucial role in maintaining the integrity of the cell envelope [19,20]. Once its structure is destroyed, the resulting stresses on membranes can be reflected in the structural properties of the bacterial envelope [24,25]. In this study, we measured the permeability and surface hydrophobicity of the outer membrane and observed the cell surface structure by using SEM to measure the changes in membrane properties caused by *waaF* deletion in *Y. enterocolitica*. As shown in Figure 2A, the hydrophobicity value of Δ*waaF* (32.6%) was significantly higher than that of the wild-type strain (16.1%), while the hydrophobicity value of Δ*waaF*-PBAD*waaF* returned to a level similar to that of the wild-type strain. Furthermore, the outer membrane permeability of Δ*waaF* was 1.48 times that of the wild-type strain, but the outer membrane permeability was restored when the mutant was complemented with the wild-type *waaF* gene (Figure 2B). As a result, the cell surface hydrophobicity and outer membrane permeability were increased in the Δ*waaF* mutant strain, indicating that the cell envelope stress was induced in response to LPS truncation. In addition, SEM results showed that the outer membrane surface of the Δ*waaF* strain had no obvious folds, holes, and other serious membrane damage, while the outer membrane surface of Δ*waaF* was relatively smooth when compared to the wild-type strain (Figure 2C).

### 2.3. Deletion of waaF Gene Increased Susceptibility of Y. enterocolitica to Environmental Stress

LPS is a physical barrier for interactions between bacteria and the environment, providing bacteria with great resistance to environmental damage and toxic molecules [22,29,30]. To test how defects in LPS structure caused by the lack of *waaF* affected the resistance to different environmental stresses, the cell growth of wild-type, Δ*waaF*, and Δ*waaF*-PBAD*waaF* strains exposed to the cationic antimicrobial peptide polymyxin B and anionic surfactant SDS was measured in this study. It should be emphasized here that the lack of *waaF* does not affect the growth of *Y. enterocolitica* in the absence of environmental stresses (Figure 3A). As shown in Figure 3B, the cell growth of the Δ*waaF* strain was significantly inhibited in the presence of 0.4 μg polymyxin B/mL. After 8 h of incubation, the cell density of the Δ*waaF* strain was decreased by 54% compared to that of the wild-type strain. However, the growth of the Δ*waaF*-PBAD*waaF* strain was close to that of the wild-type strain under polymyxin B stress. Similarly, after the addition of 0.05% SDS prior to the growth of the bacteria to the logarithmic growth phase, the growth of the Δ*waaF* strain was seriously damaged and slowly recovered in the subsequent cultivation. However, the Δ*waaF* mutant had the ability to grow under 0.05% SDS stress, but the growth ability was significantly weaker than that of the wild-type strain. The cell density of the Δ*waaF* strain decreased by 86% after 8 h of incubation compared to that of the wild-type strain. However, the growth of the Δ*waaF*–PBAD*waaF* strain with 0.05% SDS recovered closely to that of the wild-type strain (Figure 3C). These results suggest that LPS truncation caused by *waaF* deletion increased susceptibility to environmental stress in *Y. enterocolitica*.

### 2.4. Disruption of the Rcs Phosphorelay Reversed the Decreased Motility and Biofilm Formation in ΔwaaF Strain

In addition to decreased envelope stability and environmental tolerance, the LPS truncation in *Enterobacterales* also caused abnormal phenotypic changes, including flagella synthesis, biofilm formation, virulence, and pathogenicity [26,31,32,33]. In this study, swimming motility and biofilm formation were analyzed to investigate the effect of LPS truncation on the phenotypes of *Y. enterocolitica*. As shown in Figure 4, the Δ*waaF* mutant decreased the swim diameter by 53% in LBNS (LB without salts) agar, but motility was restored when the mutant was complemented with the wild-type *waaF* gene. Similarly, the defect of LPS in the Δ*waaF* strain resulted in a 62% reduction in biofilm formation, while the Δ*waaF*-PBAD*waaF* strain restored biofilm formation after 48 h of incubation, as evidenced by crystal violet staining when compared to the wild-type strain (Figure 5). These results indicate that LPS truncation caused by *waaF* mutation decreased motility and biofilm formation in *Y. enterocolitica*.

It has been reported that activation of the Rcs phosphorelay triggered by a structural deficiency of LPS suppressed the colanic acid production of *E. coli*, but the effects were reversed after the deletion of any of the Rcs phosphorelay genes [23]. In this study, Δ*waaF*-Δ*rcsF*, Δ*waaF*-Δ*rcsC*, and Δ*waaF*-Δ*rcsB* double mutants were constructed to determine whether the disruption of Rcs phosphorelay could reverse the effects caused by *waaF* deletion in *Y. enterocolitica*. As a result, the double mutants restored both swim motility and biofilm formation. The Δ*waaF*-Δ*rcsF*, Δ*waaF*-Δ*rcsC*, and Δ*waaF*-Δ*rcsB* double mutants increased the swim diameter by 64, 55, and 80%, respectively, compared to the Δ*waaF* strain (Figure 4). Similarly, biofilm formation in Δ*waaF*-Δ*rcsF*, Δ*waaF*-Δ*rcsC*, and Δ*waaF*-Δ*rcsB* double mutants showed 2.5-, 2.4-, and 2.9-fold increases, respectively, compared to the Δ*waaF* strain after 48 h of incubation (Figure 5). In addition, the swim diameter and biofilm formation of Δ*rcsF*, Δ*rcsC*, and Δ*rcsB* single mutants were also performed to distinguish the effects of LPS and Rcs phosphorelay on the motility and biofilm formation of *Y. enterocolitica*. The Δ*rcsF*, Δ*rcsC*, and Δ*rcsB* single mutants increased the swim diameter by 10, 3, and 15%, respectively, compared to the wild-type strain (Figure 4). The biofilm formation in Δ*rcsF*, Δ*rcsC*, and Δ*rcsB* single mutants was increased by 22, 14, and 38%, respectively, compared to the wild-type strain (Figure 5). Although the swim diameter and biofilm formation of Δ*waaF*-Δ*rcsF*, Δ*waaF*-Δ*rcsC*, and Δ*waaF*-Δ*rcsB* double mutants did not recover to the level of the Δ*rcsF*, Δ*rcsC*, and Δ*rcsB* single mutants, it showed that disruption of the Rcs phosphorelay in the Δ*waaF* mutant restored motility and biofilm formation to a certain extent in *Y. enterocolitica*.

### 2.5. LPS Truncation Caused by waaF Deletion Activated the Rcs Phosphorelay in Y. enterocolitica

Our previous data demonstrated that the Rcs phosphorelay represses *flhDC*, *hmsHFRS*, and *hmsT* in *Y. enterocolitica* [10,34]. The *flhDC* operon encodes the master regulator of flagella biosynthesis [10]. *hmsHFRS* and *hmsT* are genes involved in biofilm formation, of which *hmsHFRS* required for the biosynthesis of poly-β-1,6-*N*-acetylglucosamine exopolysaccharide and *hmsT* encoding diguanylate cyclase is essential for cyclic dimeric guanosine monophosphate (c-di-GMP) biosynthesis [35]. In this study, we analyzed the transcription levels of these genes to determine whether LPS truncation activated the Rcs phosphorelay in *Y. enterocolitica*. The results revealed that all these genes were repressed in the Δ*waaF* mutant strain. As a result, the transcription of *flhDC*, *hmsHFRS*, and *hmsT* in the *ΔwaaF* strain decreased by 49, 32, and 28%, respectively, while transcription was restored in the Δ*waaF*-PBAD*waaF* strain relative to the wild-type strain (Figure 6).

However, the expression levels of *flhDC*, *hmsHFRS*, and *hmsT* genes downregulated by LPS truncation were increased in all double mutant strains. The transcription of *flhDC*/*hmsHFRS*/*hmsT* in Δ*waaF*-Δ*rcsF*, Δ*waaF*-Δ*rcsC*, and Δ*waaF*-Δ*rcsB* was 1.36-times/1.38-times/1.62-times, 1.19-times/1.30-times/1.52-times, and 1.47-times/1.62-times/1.80-times that of the wild-type strain, respectively (Figure 6). The deletion of *rcsF/rcsC/rcsB* also resulted in an increased expression of these genes. However, there was no significant change between Δ*waaF*-Δ*rcsF*, Δ*waaF*-Δ*rcsC*, and Δ*waaF*-Δ*rcsB* double mutants and Δ*rcsF*, Δ*rcsC*, and Δ*rcsB* single mutants in the transcription of *flhDC*, *hmsHFRS*, and *hmsT* (Figure 6). It can be seen that *waaF* deletion did not affect the expression levels of these genes in the absence of Rcs phosphorelay in *Y. enterocolitica*. All these results provide evidence that LPS truncation caused by *waaF* deletion decreased *Y. enterocolitica* motility and biofilm formation by acting on the Rcs phosphorelay.

## 3. Discussion

LPS is the major molecular component of the outer membrane of Gram-negative bacteria and serves as a physical barrier providing the bacteria protection from its surroundings [13,21,29,30]. Among them, saturated acyl chains and hydrophilic lateral interactions between LPS molecules bridged by divalent cations make the outer membrane impermeable to both large hydrophilic molecules and hydrophobic molecules [13,36]. In this study, the deletion of the *waaF* gene failed the synthesis of the encoded heptyl glycosyltransferase II in *Y. enterocolitica* [23], thus leading to a truncated LPS structure (Figure 1B). Theoretically, the mutant LPS has only lipids and a part of the inner polysaccharide, but no outer core and O antigen [22], so it appears as a deep rough LPS (Figure 1A). The defects in the LPS structure may cause damage to the hydrophilic lateral interaction between LPS and ultimately lead to an increase in membrane permeability in *Y. enterocolitica*. In addition, the intact LPS showed a gradual increase in hydrophobicity from the inside out in terms of structural composition [20], which may explain why the LPS truncation led to increased hydrophobicity in *Y. enterocolitica*. It is known that the adhesive capacity closely related to hydrophobicity is important for bacteria to exert their toxic effects [37]. The hydrophobicity of bacteria usually alters when the cell membrane is compromised [37]. In addition to the LPS synthesis defects, some antimicrobial substances that target cell membrane, such as cationic antimicrobial peptide polymyxin B and 3,6-O-[*N*-(2-aminoethyl)-acetamide-yl]-chitosan (AACS), can also cause changes to the cell surface hydrophobicity [37,38]. In general, the truncated LPS structure caused by the deletion of *waaF* resulted in changes in membrane properties, which eventually led to the generation of envelope stress in *Y. enterocolitica*.

A new finding of this research is that the deletion of the *Y. enterocolitica waaF* gene renders the outer membrane smoother than the wild-type strain. We thought that the slightly uneven surface of the bacteria observed in the wild type and the Δ*waaF*-PBAD*waaF* strains might be due to the intact structure of LPS. LPS molecules cover more than 70% of the surface area of the outer membrane of the bacteria [21,22]. One end of its lipid A is anchored to the phospholipids of the outer membrane [39], while the sugar chain composed of core polysaccharides and O-antigen extends freely outside the membrane, endowing the outer membrane of the cell with better ductility so that a rough shape similar to villi is observed through SEM (Figure 2C). However, Δ*waaF* basically has no polysaccharide chain protrusions extending outside the phospholipids, and the remaining structure of LPS has lost its good ductility. The truncated LPS directly exposed the outer membrane to the external environment without the protection of the intact LPS; therefore, a smooth outer membrane was observed under an SEM (Figure 2C). In addition, changes in the outer membrane permeability in Δ*waaF* cells may result in the swelling of the outer membrane during the fixation and drying (sample processing for SEM analysis), thus leading to a smooth outer membrane observed through SEM.

*Y. enterocolitica* is exposed to a range of membrane stresses due to both their environmental and infectious nature, and resistance to such stress is an important property for microbial survival and virulence exertion [9,10,13]. The outer membrane of Gram-negative bacteria acts as a protective barrier at the frontline of interaction between bacteria and the environment [13], and the LPS on the outer membrane serves as a physical barrier encountered by toxic molecules and antimicrobials, preventing the destruction of cell membranes by these molecules [21,22,29,30]. In agreement with previous reports with *Y. enterocolitica* and similar findings with *E. coli*, LPS defects led to the decreased resistance to SDS and polymyxin B [22,40]. As we mentioned above, the lateral interaction between LPS prevents macromolecules from penetrating the outer membrane, providing bacteria with great resistance to toxic molecules [13,36]. Furthermore, in Gram-negative bacteria, the known polymyxin B resistance mechanisms involve outer membrane modifications and specifically those in the LPS molecule [41]. In this study, the destruction of the LPS structure and the direct exposure of the outer membrane to the environment due to the lack of protection of the complete LPS structure both can be attributed to the increase in the sensitivity of *Y. enterocolitica* to polymyxin B and SDS. These results indicate that LPS plays an important role in the survival of *Y. enterocolitica* in vitro or during the infection of animal tissues.

The Rcs phosphorelay system is an important signal transduction pathway found in many members of the *Enterobacterales* family [11,12]. This system can integrate environmental signals, regulate gene expression, and alter the physiological behavior of bacteria [42,43,44,45,46]. The outer membrane protein RcsF can sense most of the envelope stress signals that activate the activity of RcsC, trigger the downstream signal transmission of the Rcs system, RcsC→RcsD→RcsB, and finally regulate the transcription of the target gene [11,12,15]. It is reported that *E. coli* cells can activate colanic acid production through the Rcs phosphorelay system in response to a truncation of LPS [23]. Konovalova and colleagues proposed a model to explain how RcsF senses LPS defects [24,25]. In their model, RcsF forms a complex with the β-barrel and uses its positively charged, surface-exposed N-terminal domain to directly sense the state of LPS lateral interactions, thereby regulating the activity of the Rcs system [24,25]. When LPS lateral interactions are perturbed by biosynthesis defect, the RcsF N-terminus will be activated; then, the Rcs system will be triggered [24,25].

In *Y. enterocolitica*, the Rcs system was reported to sense cell envelope stress and regulate more than 130 genes involved in antibiotic resistance, bacterial chemotaxis, motility, and biofilm formation [8,10]. However, whether it can sense LPS truncation has not been reported previously. In this study, we found that the *flhDC*, *hmsHFRS*, and *hmsT* genes regulated by Rcs phosphorelay were repressed by LPS truncation, but the deletion of *rcsF*, *rcsC,* and *rcsB*, which encode Rcs phosphorelay components in Δ*waaF* cells, restored the expression levels of *flhDC*, *hmsHFRS*, and *hmsT* as well as cell motility and biofilm formation in *Y. enterocolitica*. In fact, increased *flhDC*, *hmsHFRS*, and *hmsT* were also observed in Δ*rcsF*, Δ*rcsC*, and Δ*rcsB* single mutants. However, there was no significant difference between Δ*waaF*-Δ*rcsF*, Δ*waaF*-Δ*rcsC*, and Δ*waaF*-Δ*rcsB* double mutants and Δ*rcsF*, Δ*rcsC*, and Δ*rcsB* single mutants in terms of the expression level of all these genes, suggesting that LPS truncation could not affect the expression of these genes in the inactivation of Rcs phosphorelay in *Y. enterocolitica*. A similar effect was also found in the Δ*opgGH* mutant reported by our previous study [10]. The expression of *flhDC*, *hmsT*, and *hmsHFRS* was downregulated by 45, 33, and 34%, respectively, due to the *opgGH* (encoding enzymes for synthesizing osmoregulated periplasmic glucans) deletion [10]. In this study, the transcription of *flhDC*, *hmsT*, and *hmsHFRS* was decreased by 49, 28, and 32%, respectively, which is caused by *waaF* deletion. Judging from the expression levels of these Rcs-regulated genes, the lack of *waaF* and *opgGH* has little difference in the activation degree of the Rcs system. It should be noted that LPS truncation did not affect the expression of these genes in the absence of RcsF, indicating that the RcsF protein is required for the perception of LPS defects caused by *waaF* deletion in *Y. enterocolitica*. All these data provide evidence that the truncated LPS in *Y. enterocolitica* Δ*waaF* cells might be sensed by RcsF and then activate the Rcs system, leading to decreased cell motility and biofilm formation. In addition, Bengoechea and colleagues found that the LPS O-antigen of *Y. enterocolitica* is required for virulence and the absence of O-antigen also affects the expression of other *Y. enterocolitica* virulence factors [47]; furthermore, the overexpression of O-antigen causes membrane stress that activates the CpxAR two-component signal transduction system [48]. Together, these findings suggest that the defects of LPS in the outer membrane of *Y. enterocolitica* either directly or indirectly, for example through a cellular or envelope stress, could act as a regulatory signal.

## 4. Materials and Methods

### 4.1. Bacterial Strains, Plasmids, and Culture Conditions

The bacterial strains and plasmids used in this study are listed in Table 1. *E. coli* DH5a, used as the host bacteria in plasmid construction, was cultured at 37 °C in LB medium (5 g/L yeast extract, 10 g/L tryptone, and 10 g/L NaCl). *Y. enterocolitica* ATCC23715 (biotype 1B and serotype O:8) was used as the parent strain for constructing *Y. enterocolitica* mutants. If not stated otherwise, *Y. enterocolitica* strains were grown in LB medium or LB agar plates at 26 °C. Ampicillin (100 µg/mL), chloramphenicol (15 μg/mL), and CIN (for screening *Y. enterocolitica*) composed of cefsulodin (15 µg/mL), irgasan (4 µg/mL), and novobiocin (2.5 µg/mL) were added as required.

### 4.2. Plasmids and Strains Construction

To construct pDS132-Δ*waaF*, fragments upstream and downstream to the *waaF* gene were amplified from the *Y. enterocolitica* genome using primers *waaF-up-F*/*waaF*-*up-R* and *waaF-down-F*/*waaF-down-R* (Table 2). The upstream and downstream fragments were fused and amplified by fusion PCR with primers *waaF-up-F*/*waaF*-*down-R* (Table 2). The resultant long fragment was digested with *Sph*I and *Sac*I and ligated into pDS132 digested with the same enzymes to yield pDS132-Δ*waaF*. Plasmid pDS132-Δ*waaF* was introduced into the donor strain *E. coli* S17-1 λ*pir* by electroporation and then transferred into wild-type *Y. enterocolitica* ATCC23715 by conjugation. The strategy for constructing the Δ*waaF* mutant was based on the two-step homologous recombination with plasmid pDS132 containing the *sacB* counter-selectable marker and a chloramphenicol resistant marker as described previously [10]. Similarly, plasmids pDS132-Δ*rcsF*, pDS132-Δ*rcsC*, and pDS132-Δ*rcsB* were used to construct the Δ*waaF*-Δ*rcsF*, Δ*waaF*-Δ*rcsC*, and Δ*waaF*-Δ*rcsB* double mutants, respectively.

To construct pBAD24-*waaF*, the *waaF* fragment was amplified from the *Y. enterocolitica* genome using primers *p-waaF-F*/*p-waaF-R* (Table 2), digested with *Sal*I and *Hin*dIII, and inserted into pBAD24 digested with the same enzymes. The pBAD24-*waaF* plasmid was used to transform the Δ*waaF* mutant by electroporation to yield the Δ*waaF*-PBAD*waaF* complemented strain. Ampicillin (100 µg/mL) and L-arabinose (0.6 g/L) were added to maintain plasmid pBAD24-*waaF* and induce *waaF* expression, respectively, in the Δ*waaF*-PBAD*waaF* strain. All primers used in this study are listed in Table 2.

### 4.3. Observation of Colony Morphology

The wild-type, Δ*waaF* mutant, and Δ*waaF*-PBAD*waaF* strains stored at −80 °C were streaked on LB agar plates. After the strain was cultured at 26 °C for 48 h, the morphology of the bacterial colony was observed and photographed with a digital camera.

### 4.4. LPS Isolation and Silver Stained SDS-PAGE

LPS from the wild type, Δ*waaF* mutant, and Δ*waaF*-PBAD*waaF* strains were isolated using an LPS extraction kit (iNtRON Biotechnology, Gyeonggi-do, Korea) according to the manufacturer’s instructions. The LPS samples were separated by SDS-PAGE using 8–20% gradient gels (Solarbio, Beijing, China). LPS was stained using the PAGE gel silver staining kit (Solarbio, Beijing, China) according to the manufacturer’s instructions. Gels were visualized under the white light of GelDoc-It^2^ (UVP, Upland, CA, USA).

### 4.5. Hydrophobicity Test

Cell surface hydrophobicity was determined by microbial adhesion to hydrocarbons with slight modifications [49]. Hexadecane was chosen as the non-polar solvent to reflect bacterial surface hydrophobicity. Briefly, logarithmic phase bacteria were harvested by centrifugation at 5000*× g* for 10 min, washed three times, and resuspended in phosphate buffered solution (PBS) (pH = 7.4), and the bacterial concentration was adjusted to OD_600_ = 0.5 (A_0_). The two-phase system was thoroughly mixed by vortexing for 2 min after 300 μL of hexadecane was added to 1.2 mL bacterial suspension. The aqueous phase was removed after 30 min of incubation at 37 °C to measure the absorbance at 600 nm (A_1_). The hydrophobic rate was calculated using the following formula: hydrophobic rate (%) = $\frac{\left(A_{0}-A_{1}\right)}{A_{0}}\times 100$, where A_0_ is the initial absorption value at 600 nm and A_1_ is the absorption value at 600 nm of bacteria treated with hexadecane.

### 4.6. NPN Uptake Assays

The outer membrane permeability of *Y. enterocolitica* strains was determined using the N-phenyl-1-naphthylamine (NPN) method, as described previously [37]. NPN is a hydrophobic fluorescent probe that can penetrate into the outer membrane of bacteria. This kind of probe emits a strong fluorescent signal in a hydrophobic environment (i.e., cell envelope), but a weak fluorescent signal in an aqueous environment. Therefore, the fluorescence intensity of NPN can be used to reflect the outer membrane permeability of bacteria. The procedure for NPN determination was as follows: bacteria were harvested in log phase of growth by centrifugation at 5000*× g* for 10 min, rinsed three times, and resuspended in PBS buffer (pH = 7.4) to OD_600_ = 0.5. An aliquot of 1.98 mL of bacterial suspension was fully mixed with 20 μL NPN (1 mM) and then incubated for 30 s; then, the fluorescence value was measured immediately using a fluorescence spectrophotometer (RF-5301PC, Shimadzu, Japan). Excitation and emission wavelengths for NPN were set at 350 and 429 nm, respectively, with slit widths of 5 nm.

### 4.7. Stress Survival Assays

Growth curves of *Y. enterocolitica* strains with environmental stresses were performed as described previously [22]. Strains were grown overnight at 26 °C and then diluted in 20 mL LB medium to an OD_600_ of ~0.05. To study the effects of sodium dodecyl sulfate (SDS) on the strains, cells were incubated for 2 h and 0.05% SDS was added. For the polymyxin B stress assays, a specific concentration of polymyxin B was added to the LB medium to obtain final concentrations of 0.4 μg polymyxin B/mL. Samples were taken every 2 h after inoculation, which were stored under 26 °C, 180 rpm culture conditions, and OD_600_ was determined by a spectrometer. Growth curves were plotted according to the determined OD_600_ and sampling time.

### 4.8. Motility Assays

Swimming motility assays were performed on LBNS plates containing 0.35% agar as described previously [10]. The overnight activated bacteria were transferred and cultured in the logarithmic phase, and the bacterial concentration was adjusted to OD_600_ = 1.0. Then, 1 μL bacterial suspensions were inoculated into the center of the swimming plate and incubated at 26 °C. The diameters of the swimming rings were measured using a Vernier caliper after 48 h of incubation.

### 4.9. Biofilm Assays

Crystalline violet staining was used to measure biofilm formation as described previously [10]. Logarithmic phase bacteria cultured in LNNS liquid medium were added to a 96-well plate so that each well contained 200 μL suspensions (initial OD_600_ about 0.05). The suspensions were incubated at 26 °C for 48 h and were replaced with fresh medium every 24 h. After the bacterial growth medium was removed, the biofilm was washed twice with PBS and fixed with methanol. Crystal violet staining solution (0.1%) and 33% acetic acid were used to stain the biofilm and release the dye absorbed in the biofilm, respectively, which was then measured at 595 nm.

### 4.10. Electron Microscopy of Cell Morphology

Scanning electron microscopy (SEM) was used to visualize the damage to the outer membrane as described previously [50]. Bacteria grown to the mid-log phase in LB medium were harvested by centrifugation at 5000*× g* for 10 min, washed three times, and resuspended in PBS buffer solution (pH = 7.4). The suspension was premixed with an equal volume of 2.5% glutaraldehyde for 4 h at 4 °C and subsequently dehydrated with 25, 50, 70, 80, 95, and 100% ethanol. The dehydrated samples were air-dried immediately, followed by smearing on SEM stubs and gold covering. The micrographs of the envelope were obtained using a SEM (Hitachi SU8010, Tokyo, Japan).

### 4.11. RNA Isolation and Quantitative Real-Time PCR (RT-qPCR)

Total RNA was extracted from the logarithmic phase bacteria in LNNS medium using the TransZol Up Plus RNA Kit (TransGen, Beijing, China). The RT-qPCR kit (TransGen, Beijing, China) used in this experiment reverse-transcribed RNA into cDNA in the presence of random primers, and cDNA amplification was completed in one step in the same reaction system. RT-qPCR was carried out using SYBR Green and the specific primers listed in Table 3 in a Light Cycler 480 II (Roche, Basel, Switzerland) as described previously [8,10]. Reactions were performed in triplicate, and the 16S rRNA gene was used as a reference for normalization. Relative transcription levels of the target genes were analyzed by the 2^−ΔΔCt^ method as described previously [51].

### 4.12. Statistical Analysis

All experiments were conducted in triplicate, and the results are expressed as mean ± SD. One-way analysis of variance was performed in SPSS for Windows 20.0 (SPSS Inc., Chicago, IL, USA).

## 5. Conclusions

This study mainly focused on the effect of LPS integrity in maintaining membrane function and the response regulation of the Rcs phosphorelay system to the truncated LPS in *Y. enterocolitica*. The deletion of the *Y. enterocolitica waaF* gene truncated the structure of LPS and produced deep rough LPS. This truncated LPS resulted in changes in cell surface hydrophobicity and outer membrane permeability, which in turn induced the generation of cell envelope stress. LPS truncation also led to a change in the smoothness of the membrane surface and increased susceptibility to environmental stress. The truncated LPS decreased *Y. enterocolitica* motility and biofilm formation, and this effect was reversed by disruption of the Rcs phosphorelay. Gene expression analysis indicated that Rcs phosphorelay responds to cell envelope stress induced by truncated LPS in *Y. enterocolitica*. This study reveals the importance of LPS integrity in maintaining membrane function, broadens the understanding of the Rcs phosphorelay system in response to envelope stress, and provides a theoretical basis for the development of bacterial control, prevention, and treatment.

## Figures and Tables

**Figure 1 molecules-25-05718-f001:**
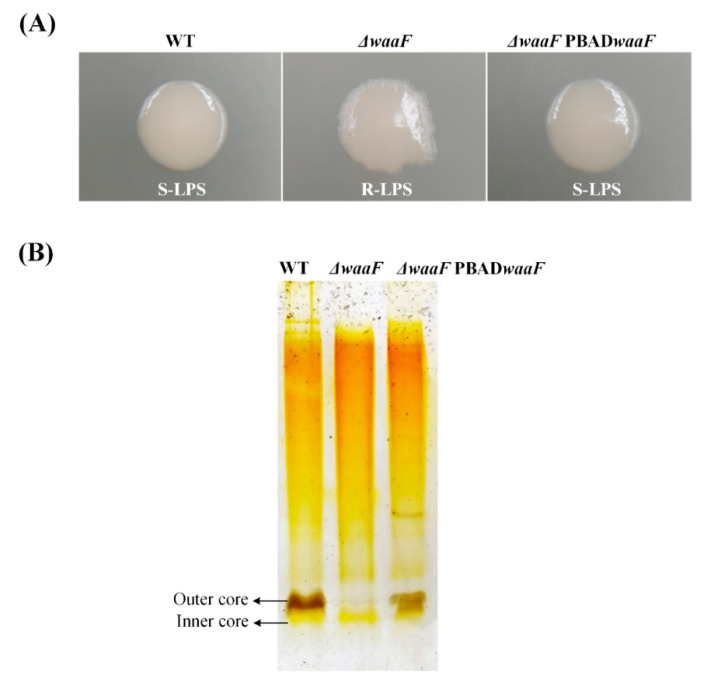
Deep rough lipopolysaccharide (LPS) was constructed by a truncated LPS structure by knocking out the *waaF* gene in *Y. enterocolitica*. (**A**) Colony morphology was transformed from smooth to rough due to the deletion of the *waaF* gene; (**B**) A truncated LPS structure was constructed by knocking out the *waaF* gene through silver-stained SDS-PAGE analysis. Representative images from three independent experiments.

**Figure 2 molecules-25-05718-f002:**
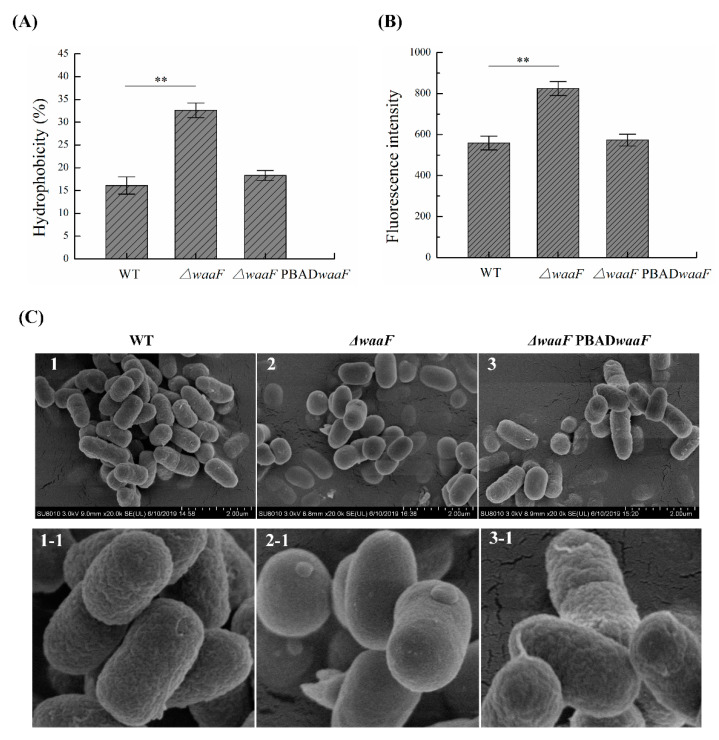
Changes in membrane properties in the wild type, Δ*waaF*, and *ΔwaaF*-PBAD*waaF* strains of *Y. enterocolitica*. (**A**) Changes in cell surface hydrophobicity; (**B**) Changes in outer membrane permeability; (**C**) Scanning electron microscope images of *Y. enterocolitica* strains. Among them, sub-figures (**1-1**, **2-1**, **3-1**) are partial magnified (100 times) views of sub-figures (**1**–**3**), respectively. Data are average values and standard deviations of triplicate experiments. An asterisk indicates a significant difference with ** *p* < 0.01.

**Figure 3 molecules-25-05718-f003:**
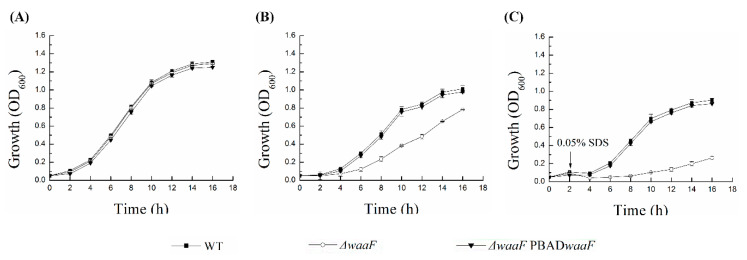
Effect of environmental stresses on the wild type, Δ*waaF*, and Δ*waaF*-PBAD*waaF* strains of *Y. enterocolitica*. (**A**) Growth curves of *Y. enterocolitica* strains without environmental stresses; (**B**) Growth curves of *Y. enterocolitica* strains in the presence of 0.4 μg polymyxin B/mL; (**C**) Growth curves of *Y. enterocolitica* strains in the presence of 0.05% SDS. The starter cultures were incubated in Luria Broth (LB) medium supplemented with 0.6 g/L L-arabinose at 26 °C. Cell growth was measured every 2 h at 600 nm in a spectrophotometer. Data are average values and standard deviations of triplicate experiments.

**Figure 4 molecules-25-05718-f004:**
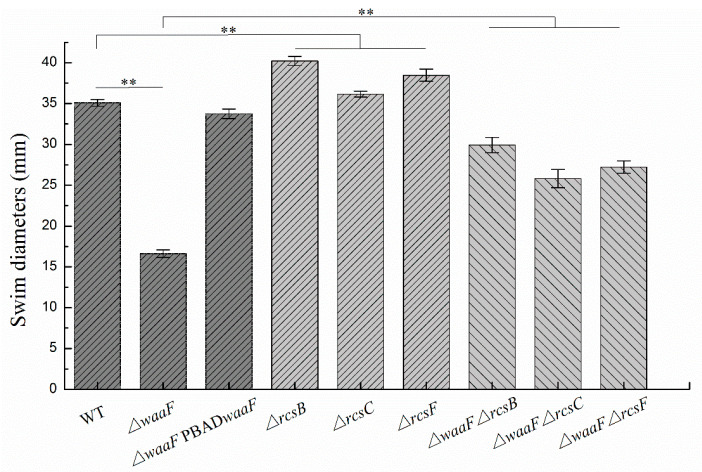
Changes in swim motility of wild-type and mutant strains of *Y. enterocolitica*. *Y. enterocolitica* was grown in LBNS semisolid plates supplemented with 0.6 g/L L-arabinose at 26 °C. Swim diameters were measured after 48 h of incubation. Data are average values and standard deviations of triplicate experiments. The statistical comparison shows that there is a significant difference between WT (wild-type) and Δ*waaF*; there is a significant difference between WT and Δ*rcsF*/Δ*rcsC*/Δ*rcsB* single mutant; there is a significant difference between Δ*waaF* and Δ*waaF*-Δ*rcsF*/Δ*waaF*-Δ*rcsC*/Δ*waaF*-Δ*rcsB* double mutant. An asterisk indicates a significant difference with ** *p* < 0.01.

**Figure 5 molecules-25-05718-f005:**
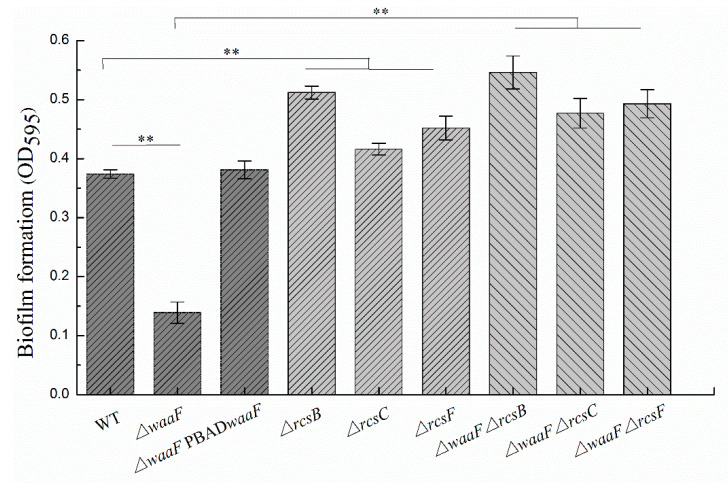
Changes in biofilm formation of wild-type and mutant strains of *Y. enterocolitica*. *Y. enterocolitica* was cultured in LNNS (LBNS broth diluted 1:8 in distilled water) medium supplemented with 0.6 g/L L-arabinose at 26 °C. Biofilm formation was analyzed after 48 h of incubation by staining with crystal violet and measuring absorbance at 595 nm. Data are average values and standard deviations of triplicate experiments. The statistical comparison shows that there is a significant difference between WT and Δ*waaF*; there is a significant difference between WT and Δ*rcsF*/Δ*rcsC*/Δ*rcsB* single mutant; there is a significant difference between Δ*waaF* and Δ*waaF*-Δ*rcsF*/Δ*waaF*-Δ*rcsC*/Δ*waaF*-Δ*rcsB* double mutant. An asterisk indicates a significant difference with ** *p* < 0.01.

**Figure 6 molecules-25-05718-f006:**
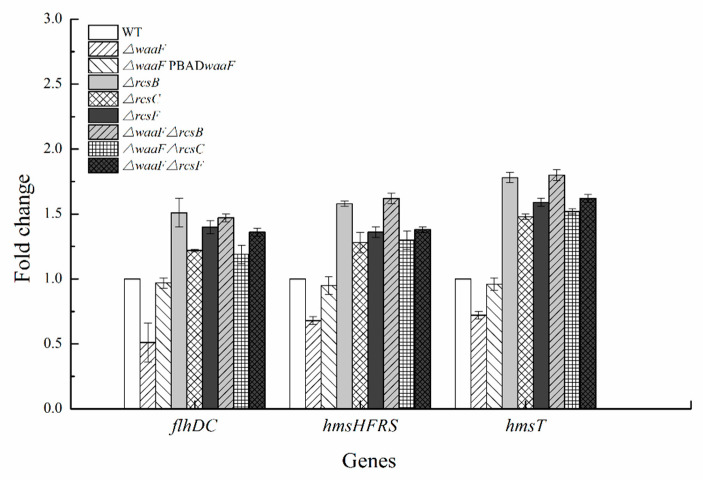
Transcriptional changes in the mutant strains of *Y. enterocolitica*. *Y. enterocolitica* was grown to the mid-log phase in LNNS medium, and total RNA was extracted. Expression of *flhDC*, *hmsHFRS*, and *hmsT* was determined by RT-qPCR in vitro. The 16S rRNA gene was used as a normalization control. Data are average values and standard deviations of triplicate experiments.

**Table 1 molecules-25-05718-t001:** Bacterial strains and plasmids used in this study.

Strains and Plasmids	Relevant Characteristics	Sources
***Y. enterocolitica***		
YE	WT, Serotype O:8, Biotype 1B, pYV-	[34]
YE-W	Δ*waaF*	This study
YE-W+	Δ*waaF*, PBAD*waaF*; Ampr	This study
YE-F	Δ*rcsF*	[10]
YE-C	Δ*rcsC*	[10]
YE-B	Δ*rcsB*	[34]
YE-WF	Δ*waaF*, Δ*rcsF*	This study
YE-WC	Δ*waaF*, Δ*rcsC*	This study
YE-WB	Δ*waaF*, Δ*rcsB*	This study
***E. coli***		
S17-1 λ*pir*	*recA1*, *thi*, *pro*, *hsdR*-M+, RP4:2-Tc::Mu-Kan::Tn*7*, λpir	[34]
DH5a	F-, φ80*lacZ*Δ*M15*, Δ(*lacZYA-argF*)*U169*, *deoR*, *recA1*, *endA1*, *hsdR17*(rk−, mk+), *phoA*, *supE44*, *λ*-, *thi-1*, *gyrA96*, *relA1*	[34]
**Plasmids**		
pDS132	conditional replication vector; R6K origin, mobRK4 transfer origin, sucrose-inducible-*sacB*; Cmr	[34]
pDS132-Δ*waaF*	upstream and downstream fragments of *waaF* gene were cloned into pDS132; Cmr	This study
pDS132-Δ*rcsF*	upstream and downstream fragments of *rcsF* gene were cloned into pDS132; Cmr	[10]
pDS132-Δ*rcsC*	upstream and downstream fragments of *rcsC* gene were cloned into pDS132; Cmr	[10]
pDS132-Δ*rcsB*	upstream and downstream fragments of *rcsB* gene were cloned into pDS132; Cmr	[34]
pBAD24	*AraC*, promoter PBAD; Ampr	[34]
pBAD24-*waaF*	*AraC*, PBAD-*waaF*; Ampr	This study

*Amp*, ampicillin; *Cm*, chloramphenicol; *r*, resistance.

**Table 2 molecules-25-05718-t002:** Primers used for strain and plasmid construction in this study.

Primer	Sequences (5′→3′)
*waaF-up-F*	ATCGCATGCTGCCACAAGCTGATTCACAGA
*waaF-up-R*	ACCGTTTATCAATTCCTTGCAGCAAGTTATT
*waaF-down-F*	AATAACTTGCTGCAAGGAATTGATAAACGGTTGCATGTATTGATCGTTAAAA
*waaF-down-R*	ATTCAGAGCTCCTGCGCAATAGCATAATCGCC
*p-waaF-F*	ATGCGTCGACATGAAAATACTGGTAATCG
*p-waaF-R*	CCCAAGCTTTTAATCGCCCTCTTTCACA
*rcsF-up-F*	ACTGCATGCAAATCATTGGAAGAACTGCAAC
*rcsF-down-R*	ACTGCGAGCTCCTTTGCGGTAGGCTGGGCGTG
*rcsC-up-F*	ACTGCATGCCTCAATGGCGACGATCGGGTTA
*rcsC-down-R*	ACTGCGAGCTCCAGATTTAGCCATAATAGTAC
*rcsB-up-F*	ACTGCATGCAGAAGTGCGTTCTATAATCACA
*rcsB-down-R*	ACTGCGAGCTCATCTGGATGAGAATGCAGATC

Restriction sites are underlined.

**Table 3 molecules-25-05718-t003:** Primers used in the RT-qPCR assay.

Primer	Sequences (5′→3′)	Amplicon Size (bp)
*q-flhDC-F*	CCTCAGCGATGTTTCGTCTC	176
*q-flhDC-R*	CTGCAAGTCATCCACACGAG	
*q-hmsHFRS-F*	GATGATGTACCGCCTCCAGA	96
*q-hmsHFRS-R*	GTGAATAGTTTCCCGCGCAT	
*q-hmsT-F*	TATAATCGCCGTGGGTTGGA	144
*q-hmsT-R*	CACTAAGGCTTGGTCTCCCA	
*16S rRNA-F*	GCACGTAATGGTGGGAACTC	183
*16S rRNA-R*	CTCCAATCCGGACTACGACA	

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
