# Peer review of "Rcs Phosphorelay Responses to Truncated Lipopolysaccharide-Induced Cell Envelope Stress in *Yersinia enterocolitica"

_molecules, 2020, doi:10.3390/molecules25235718_

Round 1

Reviewer 1 Report

In this study Meng and colleagues examined the effects of the disruption of waaF gene encoding a biosynthetic enzyme of LPS, on the structure and function of cell envelope of Yersinia enterocolitica. The authors also report that the truncated LPS in the absence of waaF gene activates the Rcs phosphorelay envelope stress response, and that activation of Rcs system is the cause of decreased cell motility and biofilm formation observed in the ΔwaaF strain of Y. enterocolitica.

The results reported in this paper can be anticipated from knowledge accumulated from earlier reports describing the effect of the waaF gene disruption on the structure of LPS, cell morphology, and activation of envelope stress responses in other enterobacteria, which lower the novelty or significance and interest to the readers. However, this study is valuable in that it is a systematic and comprehensive study on the effect of LPS truncation on the structure and function of cell envelope of Y. enterocolitica and provides additional information about bacterial cell surfaces.

I have a few question about the description in the Materials and Methods section.

Lane 313. LPS were detected using a UV transilluminator. Please provide the excitation wavelength of and the detection wavelength.

Lane 329. The authors monitored outer membrane permeability by measuring NPN fluorescence. Because an increase rate of fluorescence but not a final value of fluorescence intensity may reflect the outer membrane permeability, information about the incubation time with NPN should be provided.

Reviewer 2 Report

In this work (“Rcs phosphorelay responses to truncated lipopolysaccharide-induced cell envelope stress in Yersinia enterocolitica”), Meng and colleagues investigate the effect of LPS deep rough mutants on the functioning of the outer membrane of Y. enterocolitica and on the activation of the Rcs stress response.  The authors determine that deletion of waaF causes a deep rough LPS phenotype decreasing the function of the OM permeability barrier. The authors further demonstrate that the Rcs stress response is induced in this strain causing changes to motility and biofilm production.

A great deal of this work confirms that interactions and phenotypes that occur in E. coli also occur in Y. enterocolitica. Although this somewhat limits the novelty of the work, it will still be of interest to readers given Y. enterocolitica’s pathogenic status and the general low level of information available on the functioning of the Rcs system in Yersinia. The study is generally rigorous and scientifically sound. However, there are several concerns that, if addressed, would increase the support for the conclusions and strengthen the paper.

Major Concerns

  1. Figure 1B: This silver stain is difficult to easily see. However, it seems like there is less LPS overall in the waaF It seems like the lower deep rough band is present in all samples equally, but the complete core band disappears in the waaF mutant. The authors should quantify the amounts of these forms of LPS and comment on this discrepancy.
  2. Figure 2B: The authors say that outer membrane permeability was increased based on a measure of fluorescence intensity. It is not at all clear from the Figure or the corresponding text what is being measured here. The assay needs to be better described. In addition, different mutations can affect OM permeability to different compounds in different ways. Therefore, the authors need to be more specific as to what type of compound permeability is increased for (e.g., large, hydrophobic, etc.).
  3. Figure 3C: The authors state that the waaF mutant is damaged by SDS but then recovers and grows. This may be true but the growth pattern is quite indicative of suppressor mutant(s) growing out. The authors should take bacteria from the final culture and retreat with SDS to determine whether the cells that grow are suppressor mutations or acknowledge this possibility in the text.
  4. Figure 6: Although it is clear from this data that the Rcs response is on. It is not clear whether waaF is a strong or weak inducer from this data. A control for Rcs induction such as a mutant for osmoregulated periplasmic glycan production should be included for comparison.

Minor Comments

  1. Pg 1, Ln 41 and throughout: It is probably clearer at this point to use “Enterobacterales” in place of “Enterobacteriaceae” since in the new naming scheme Enterobacteriaceae is a family within the order of Enterobacterales (and not the one that holds Yersinia).
  2. 2, Ln 55: “defects”
  3. 2, Ln 69: “Ren and colleagues”
  4. 2, Ln 89, Figure 1: I don’t think the “S-R” abbreviation has been defined.
  5. 2, Ln 93: Why is this indirect? It seems like assaying the size of the LPS is a relatively direct way to see if the core is truncated.
  6. 3, Ln 105-106: It would be helpful to a general audience if the authors commented on the implications of surface hydrophobicity and the types of mutants and/or conditions that do and do not cause changes to surface hydrophobicity measurements here or in the discussion.
  7. Figure 3: It would be helpful to increase the font sizes in this figure.
  8. 4, Ln 132: I am not sure what “in the subsequent determination” means.
  9. Figures 4 and 5: The comparisons for statistics are somewhat confusing in these figures. I think change in waaF strains verses their parents are being compared but I am not sure. This could be shown in a clearer way and should be explained in the figure legends.
  10. 6, Ln 185: “represses” would be more clear than “regulates”
  11. 6, Ln 192: The authors state “decreased by 0.49-, 0.32-, and 0.28-fold”. When reading this I was confused as to how to interpret these numbers as a change from “1” for the wild type to 0.5-fold is actually a 2-fold decrease. This wording needs to be clarified.
  12. 7, Ln 211-215: In general, this paper is written very well. However, these two sentences are a little rough and require revision.
  13. 7, Ln 229 & 231: I am not sure exactly what you mean by ductility in this context.
  14. 7, Ln 223-233: Has this “smooth” phenotype been observed for other deep rough mutants in Y. enterocolitica or in other species?
  15. 12, Ln 394: “revision”

Reviewer 3 Report

The manuscript describes the influence of truncation of the LPS structure of Yersinia enterocolitica serotype O:8 on the functionality of the outer membrane, and the role of the Rcf system on it. In general, the manuscript in well written and interesting. I have some minor and major comments on it as listed below.

L29. the references 1-4 are not appropriate to the first sentence in the introduction. the authors should use original references, not their own articles.

L30. ref 5 is old, the authors should use a recent annual report of EFSA

L83. The Y. enterocolitica strain used in the work should be introduced here. Biotype, serotype etc.

Figure 1B. The quality of the LPS gel is not adequate. The authors should refer to the work of Bengoechea et al. Mol Microbiol 2004, vol 52, p451-469, Figure 1, where silver-stained analysis of LPS of Yersinia enterocolitica serotype O:8 strain is  shown. The presence of the characteristic O-antigen ladder should be visible in the wild type and the complemented strains.

Figure 2C. The resolution of the surface structures is not satisfactory in printed article. Only inspection of magnified images on screen allows detection of the surface differences.

L187. The correct nomenclature for sugars should be used. greek letter for beta, and  N in italics

L192. I find the usage of fold-differences in the transcription confusing, it would be better to use percentages

L214. The sentence is awkward

L223-226. As the authors have no experimental data to back up this conclusion, only the SEM images, I would not put too much weight on it. The bacteria are fixed and dried for SEM, so the permeability of the cells might be different and cause some swelling of the waaF mutant bacteria smoothing out the surface.

In general the authors should discuss their results in the light of the work carried out on LPS of Y. enterocolitica O:8 by Bengoechea et al  

  1. Bengoechea, J.A.; Najdenski, H.; Skurnik, M. Lipopolysaccharide O antigen status of Yersinia enterocolitica O:8 is essential for virulence and absence of O antigen affects the expression of other Yersinia virulence factors. Mol. Microbiol. 2004, 52, 451-469.
  2. Bengoechea, J.A.; Skurnik, M. Temperature-regulated efflux pump / potassium antiporter system mediates resistance to cationic antimicrobial peptides in Yersinia. Mol. Microbiol. 2000, 37, 67-80.
  3. Bengoechea, J.A.; Zhang, L.; Toivanen, P.; Skurnik, M. Regulatory network of lipopolysaccharide O-antigen biosynthesis in Yersinia enterocolitica includes cell envelope-dependent signals. Mol. Microbiol. 2002, 44, 1045-1062.

Table 1. The references for the bacterial strains should be given, it is not enough to write that a lab stock was used. All the strains and plasmids marked as lab stock need references

L286-287. Cite to Table 2 for the primers

L293. Similarly, plasmids pD… were used …

L298. … to yield the in trans complemented strain ….

In the methods, it would be proper to cite the original articles describing the methods. In addition sections 4.74.8, 4.9, 4.10 need references as the methods were not developed in the present manuscript.

Why is section 5. Conclusiomns placed after the Materials and methods section?

Round 2

Reviewer 3 Report

The authors have responded appropriately to my criticism